# Few-Shot Continual Active Learning by a Robot

**Ali Ayub**
University of Waterloo
Waterloo, ON N2L3G1, Canada
a9ayub@uwaterloo.ca

**Carter Fendley**
Capital One
New York, NY 10017, USA
ccf5164@psu.edu

## Abstract

In this paper, we consider a challenging but realistic continual learning problem, Few-Shot Continual Active Learning (FoCAL), where a CL agent is provided with unlabeled data for a new or a previously learned task in each increment and the agent only has limited labeling budget available. Towards this, we build on the continual learning and active learning literature and develop a framework that can allow a CL agent to continually learn new object classes from a few labeled training examples. Our framework represents each object class using a uniform Gaussian mixture model (GMM) and uses pseudo-rehearsal to mitigate catastrophic forgetting. The framework also uses uncertainty measures on the Gaussian representations of the previously learned classes to find the most informative samples to be labeled in an increment. We evaluate our approach on the CORe-50 dataset and on a real humanoid robot for the object classification task. The results show that our approach not only produces state-of-the-art results on the dataset but also allows a real robot to continually learn unseen objects in a real environment with limited labeling supervision provided by its user[1].

## 1 Introduction

Continual learning (CL) [3, 4, 5, 6] has emerged as a popular area of research in recent years because of its limitless real-world applications, such as domestic robots, autonomous cars, etc. Most continual machine learning models [7, 8, 9, 10], however, are developed for constrained task-based continual learning setups, where a CL model continually learns a sequence of tasks, one at a time, with all the data of the current task labeled and available in an increment. Real world systems, particularly autonomous robots, do not have the luxury of getting a large amount of labeled data for each task. In contrast, robots operating in real-world environments mostly have to learn from supervision provided by their users [11, 5, 12]. Human teachers, however, would be unwilling answer a large number of questions or label a large amount of data for the robot. It would therefore be useful for robots to self-supervise their learning, and ask the human teachers to label the most informative training samples from the environment, in each increment. In this paper, we focus on this challenging problem, termed as Few-Shot Continual Active Learning (FoCAL).

One of the main problems faced by continual machine learning models is catastrophic forgetting, in which the CL model forgets the previous learned tasks when learning new knowledge. In recent years, several works in CL have focused on mitigating the catastrophic forgetting problem [13, 3, 6]. Most of these works, however, are developed for the task-based continual learning setup, where the model assumes that all the data for a task is available in an increment and it is fully labeled. These constraints are costly and limit the real-world application of CL models on robots. Active learning has emerged as an area of research in recent years, where machine learning models can choose the most informative samples to be labeled from a large corpus of unlabeled data, thus reducing the labelling

---

[1]A preliminary version [1, 2] of this work was presented at workshops in RoMan 2020 and ICRA 2021.

36th Conference on Neural Information Processing Systems (NeurIPS 2022).

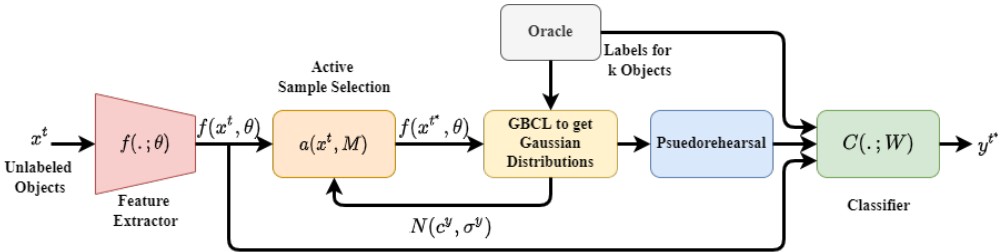

Figure 1: Our overall framework for FoCAL. In each increment $t$, the features extracted for unlabeled objects $f(x^t; \theta)$ are passed through the acquisition function $a(x^t, \mathcal{M})$ to get $k$ most informative samples $x^{t^*}$, which are labeled by the oracle. The labeled feature vectors are used to update the GMM representation of the learned classes $Y^t$. Pseudo-rehearsal is used to replay old class data, and the classifier model $C(.; W)$ is trained on the pseudo-samples of the old classes and the labeled feature vectors in the $t$th increment.

effort [14, 15]. Most active learning techniques use uncertainty sampling to request labels for the most uncertain objects [15, 14, 16]. These techniques, however, do not learn continually and thus would suffer from catastrophic forgetting. These issues related to the development of "close-world" techniques for continual learning, active learning and open-set recognition have been explored in detail in [17].

In this paper, we consider FoCAL for the online continual learning scenario for the image classification task. In this setup, a CL model (applied on a robot) receives a small amount of unlabeled image data of objects from the environment in an increment, where the objects can belong to the previously learned classes by the model, or new classes. The model is allowed to get a small number of object samples to be labeled by the user. As the model continues to learn from new training samples, it does not have access to the raw image data of the previously learned objects. Overall, FoCAL is a combination of multiple challenging problems in machine learning, mainly Few-Shot Class Incremental Learning (FSCIL) [9, 10], Active Learning [14, 18], and online continual learning [19]. To solve FoCAL, we get inspiration from the continual learning and active learning literature, to develop protocols for continual learning models so that they can actively choose informative samples in an increment. Particularly, we take inspiration from FSCIL literature to develop a new FoCAL model, in which we learn and preserve the feature representation of the previously learned objects classes by modelling them as Gaussian mixture models. To mitigate catastrophic forgetting, we use pseudo-rehearsal [20] using the samples generated from the Gaussian distributions of the old classes, thus removing the need to store raw data for the classes. Further, to choose most informative samples from an unlabeled set, we use a combination of predictive entropy [21, 18] and viewpoint consistency metrics [14, 16] on the GMM representation of the previously learned classes. We perform extensive evaluations of our proposed approach on the CORe-50 dataset [22], and on a real humanoid robot in an indoor environment. Our approach outperforms state-of-the-art (SOTA) continual learning approaches for FoCAL on the CORe-50 datast with significant margins. Further, our approach can also be integrated on a humanoid robot, and allow the robot to learn a large number of common household objects over a long period of time with limited supervision provided by the user. Finally, as a part of this work, we also release the object dataset collected by our robot as a benchmark for future evaluations for FoCAL (available here: `https://tinyurl.com/2vuwv8ye`).

## 2 Few-Shot Continual Active Learning

We define the Few-Shot Continual Active Learning (FoCAL) problem as follows: Suppose that an AI agent (e.g. a robot) gets a stream of unlabeled data sets $D^1_{pool}, D^2_{pool}, ..., D^t_{pool}, ...$ over $t$ increments, where $D^t_{pool} = \{x^t_i\}_{i=1}^{|D^t_{pool}|}$. In each increment, a continual learning model $\mathcal{M}$ with parameters $\Theta$ can only obtain a small number ($k^t < |D^t_{pool}|$) of samples to be labeled. Given the model $\mathcal{M}$, an acquisition function $a(x^t, \mathcal{M})$, where $x^t \in D^t_{pool}$, is used by the AI agent to find the most informative samples to be labeled in an increment $t$: $x^{t^*} = \mathrm{argmax}_{D^t_{pool}} a(x^t, M)$. Therefore, in each increment $t$, the model $\mathcal{M}$ gets trained on small subsets of labeled data $D^t = \{(x^{t^*}_i, y^t_i)\}_{i=1}^{|k^t|}$, where $y^t_i \in Y^t$

represents the class label of $x_i^{t^*}$ and $Y^t$ is the set of classes in the $t$-th increment. Note that unlike most continual learning setups, $Y^i \cap Y^j \neq \varnothing, \forall i \neq j$. After training on $D^t$, the model $\mathcal{M}$ is tested to recognize all the encountered classes so far $Y^1, Y^2, ..., Y^t$. The main challenges of FoCAL are three-fold: (1) avoid catastrophic forgetting, (2) prevent overfitting on the few training samples, (3) efficiently choose most informative samples in each increment.

For FoCAL for the task of object classification, we consider the model $\mathcal{M}$ (a CNN) as a composition of a feature extractor $f(.;\theta)$ with parameters $\theta$ and a classification model with weights $W$. The feature extractor transforms the input images into a feature space $\mathcal{F} \in \mathbb{R}^n$. The classification model takes the features generated by the feature extractor, and generates an output vector followed by a softmax function to generate multi-class probabilities. In this paper, we use a pre-trained feature extractor, therefore parameters $\theta$ are fixed. Thus, we incrementally finetune the classification model on $D^1, D^2, ...$ and get parameters $W^1, W^2, ....$ In an increment $t$, we expand the output layer by $|Y^t|$ neurons to incorporate new classes. Note that this setup does not alleviate the three challenges of FoCAL mentioned above. The subsections below describe the main components of our framework (Figure 1) to transform this setup for FoCAL.

## 2.1 GMM Based Continual Learning (GBCL)

We aim to develop a model that not only helps the system with continual learning but is also motivated by the newness of an object. To accomplish this, we must evaluate how different an incoming object is from previously learned object classes, ideally without any additional supervision. To accomplish this, we consider a clustering-based approach to represent the distribution of object classes. Unlike previous works on clustering-based approaches for continual learning [10, 11] that represent the object classes as mean feature vectors (centroids), we estimate the distribution of the each object class using a uniform Gaussian mixture model (GMM). We believe that representing each class data as a GMM may better represent the true distribution of the data rather than assuming that the distribution is circular. We call our complete algorithm for continually learning GMMs of multiple object classes as GMM based continual learning (GBCL).

Once the $k$ feature vectors ($D^t$) selected by acquisition function (Section 2.2) as most informative samples are labeled by the oracle in increment $t$, GBCL is applied to learn GMMs for the classes $Y^t$. For each $i$th feature vector $x_i^{t^*}$ in $D^t$ labeled as $y_i^t$, if $y_i^t$ is a new class never seen by the model before, we initialize a new Gaussian distribution $\mathcal{N}(x_i^t, O)$ for class $y$ with $x_i^t$ as the mean (centroid) and a zero matrix ($O$) as the covariance matrix[2]. Otherwise, if $y_i^t$ is a known class, we find the probabilities $\mathcal{N}(x_i^t|c_1^y, \sigma_1^y), ..., \mathcal{N}(x_i^t|c_j^y, \sigma_j^y), ..., \mathcal{N}(x_i^t|c_{n^y}^y, \sigma_{n^y}^y)$ for $x_i^y$ to belong to all the previously learned Gaussian distributions for class $y$, where $n^y$ is the total number of mixture components in the GMM for class $y$, and $c_j^y$ and $\sigma_j^y$ represent the centroid and covariance matrix for the $j$th mixture component of class $y$, respectively. If the maximum probability among the calculated probabilities for all the distributions is higher than a pre-defined probability threshold $P$, $x_i^t$ is used to update the parameters (centroid and covariance matrix) of the most probable distribution ($\mathcal{N}(c_j^y, \sigma_j^y)$) in class $y$. The updated centroid $\hat{c}_j^y$ is calculated as a weighted mean between the previous centroid $c_j^y$ and $x_i^t$:

$$\hat{c}_j^y = \frac{w_j^y \times c_j^y + x_i^t}{w_j^y + 1} \tag{1}$$

where, $w_j^y$ is the number of images already clustered in the $j$th (most probable) Gaussian distribution. The updated covariance matrix $\hat{\sigma}_j^y$ is calculated based on the procedure described in [23]):

$$\hat{\sigma}_j^y = \frac{w_j^y - 1}{w_j^y}\sigma_j^y + \frac{w_j^y - 1}{w_j^{y2}}(x_i^t - \hat{c}_j^y)^T(x_i^t - \hat{c}_j^y) \tag{2}$$

where, $\sigma_j^y$ is the previous covariance matrix and $(x_i^t - \hat{(c)}_j^y)^T(x_i^t - \hat{c}_j^y)$ represents the covariance between $x_i^t$ and $\hat{c}_j^y$. If, on the other hand, the maximum probability among the calculated probabilities

---

[2]We do not describe mixing coefficients here, as they will always be $1/n$ for a uniform GMM, where $n$ is the number of mixture components.

for all the distributions is lower than $P$, a new Gaussian distribution $\mathcal{N}(x_i^t, O)$ is created for class $y$ with $x_i^t$ as the centroid and $O$ as the covariance matrix.

The result of this process is a set of $N^t$ uniform GMMs with parameters $\phi^1, \phi^2, ..., \phi^{N_t}$ for $N^t$ classes learned up till increment $t$. Note that instead of using the number of mixture components as a hyperparameter, we use the probability threshold. This way we can start with a simple distribution model for each class (a single mixture component) and add more mixture components only when the new images of the class are too dissimilar from the previous mixture components, and thus cannot be modeled by the GMM. Therefore, the total number of mixture components for each class can be different based on the similarity between the images of the class. In section 2.2, we use the same idea of dissimilarity between an unlabeled image and a GMM to predict most informative samples.

### 2.1.1 Pseudo-rehearsal and Classifier Training

To avoid catastrophic forgetting, we use pseudo-rehearsal [20] to replay the old classes when learning from new data in increment $t$. For pseudo-rehearsal, we sample the Gaussian distributions in the GMMs of all the previously learned classes to generate a set of pseudo-feature vectors. Note that we also store the total number of images clustered in each Gaussian distribution of the classes ($w_j^y$) during the GMM learning phase (Section 2.1). Therefore, we generate the same number of pseudo-feature vectors as the original number of images for each class. After generating the pseudo-feature vectors, the classifier model $C(.; W)$ is trained using the labeled dataset $D^t$ in increment $t$, and the pseudo-feature vectors of the previous classes.

For classification of a test image $x$, it is first passed through the feature extractor $f(x, \theta)$ and then through the classifier $C(f(x, \theta), W)$. Softmax function ($\sigma$) is then applied on the output to generate class probabilities, and the class $y^*$ with the maximum probability is predicted as the label for the test image: $y^* = \mathrm{argmax}_y \, \sigma(W^T f(x, \theta))$.

## 2.2 Active Learning using GMMs

We quantify the novelty of an object in terms of how much the model is uncertain about the object. Unlike most active learning setups [18, 14], in FoCAL the model does not have access to a training set in each increment for training the model to predict uncertain object classes. Further, even if the model does get trained to predict unknown object classes in the first increment, it will catastrophically forget the criterion of novelty as it continually learns new object classes in the subsequent increments (unknown classes in the first increment become known to the model in the subsequent increments). Therefore, we do not train our model for active learning, and instead use the GMM representations of the previously learned object classes to predict the most uncertain objects.

Considering the FoCAL setup (as described in Section 2), in an increment $t$, the AI agent gets an unlabeled dataset $D_{pool}^t$ and it must find $k < |D_{pool}^t|$ most informative object samples from the dataset to be labeled. To develop an acquisition function for this, we use a combination of two active learning techniques applied to the GMM representations of the previously learned object classes.

First, we use the prediction entropy $\mathbb{H}[y^*|x_i^t]$ of an object as the acquisition function [21]:

$$\mathbb{H}[y^*|x_i^t] = - \sum_{y=1}^{N^{t-1}} p(y^* = y|x_i^t) \mathrm{log} p(y^* = y|x_i^t) \tag{3}$$

For an unlabeled data point $x_i^t \in D_{pool}^t$, we find the predictive probability of $x_i^t$ using the GMM representation of the object classes learned in the previous increments. The predictive probability $p(x_i^t|\phi^y)$ for of $x_i^t$ to belong to the GMM of a class $y$ can be defined as:

$$p(x_i^t|\phi^y) = \frac{1}{n^y} \sum_{j=1}^{n^y} \mathcal{N}(x_i^t|c_j^y, \sigma_j^y) \tag{4}$$

Intuitively, if a sample $x_i^t$ is already learned by the AI agent, then its probability to belong to one of the previously learned class GMMs must be high, and thus entropy for $x_i^t$ must be low. Therefore, top $k$ samples with the highest entropy can be chosen as the most informative samples.

For the second technique, we use the idea of viewpoint consistency used in active learning [14, 16]. The main idea is that if an object is already learned by the agent before, then consistent predictions must be produced for the object under different viewpoints (see Figure 3 in the supplementary file for an example). We again use the GMM representations of the previously learned classes for this acquisition function. Let's consider that there are multiple ($l$) viewpoints $x_1^t, ..., x_i^t, ..., x_l^t$ of an unlabeled object $j$ available to the AI agent in increment $t$. We find the predictive probability of each $x_i^t$ to belong to the GMM representations of the previously learned classes (equation (6)). Using the maximum predictive probability over all the classes we find the class prediction $y_i^{t^*}$ for each $x_i^t$ as $y_i^{t^*} = \operatorname{argmax}_{y=1,...,N^t} p(x_i^t | \phi^y)$. Next, we count the total number of times each of the $N^{t-1}$ classes is predicted among the different viewpoints of the object $j$. Let $S_j = \{s_j^1, ..., s_j^y, ..., s_j^{N^{t-1}}\}$ represent the total number of times each class is predicted among the different viewpoints of the object. We normalize set $S_j$ by dividing each $s_j^y$ by $l$ such that $\sum_{y=1}^{N^{t-1}} s_j^y = 1$. We then take the inverse of the maximum value in $S_j$ ($1/\max S_j$), which represents the inverse of the highest percentage of viewpoints of object $j$ that are predicted consistently. Thus, we use this term as the inconsistency score of the object $j$.

We use a combination of predictive entropy and viewpoint consistency to generate the final acquisition function for our framework. Before combining the two metrics, we transform the predictive entropy for multiple viewpoints of the same objects as it is originally designed for individual data samples. For multiple viewpoints of an object $j$, we find the predictive entropy of all the viewpoints individually and then take an average to get the overall predictive entropy of object $j$. Therefore, we maximize the following combined function to get the most informative samples in increment $t$:

$$\frac{\delta}{l} \sum_{m=1}^{l} \mathbb{H}[y^* | x_m^t] + (1 - \delta) \frac{1}{\max S_j} \tag{5}$$

where, $\delta$ is a hyperparameter that controls the contribution of predictive entropy and viewpoint consistency towards the overall uncertainty score of an object. Note that the viewpoint consistency can only be applied when the agent has access to multiple viewpoints of the same object. In cases when multiple viewpoints are unavailable, we can use the predictive entropy alone. In our experiments (Section 3), however, we use multiple viewpoints of all the objects.

## 3 Experiments

We first evaluate our approach for FoCAL on the Core-50 dataset [22] and then using the Pepper robot. We begin by presenting the implementation details and then compare our approach against SOTA continual learning approaches on the CORe-50 dataset. Finally, we present the evaluations of our approach on Pepper. Details about the CORe-50 dataset are in the supplementary file.

### 3.1 Implementation Details

We used the Pytorch deep learning framework [24] for implementation and training of all neural network models. We used ResNet-18 [25] pre-trained on the ImageNet dataset [26] as a feature extractor for GBCL. The same pre-trained network was also used in all the other continual learning approaches for a fair comparison. For GBCL, we only used the diagonal entries of the covariance matrix to keep the memory budget from growing drastically.

For FoCAL experiments on the CORe-50 dataset, we develop a new experimental setup, as the standard active learning and continual learning experimental setups are not sufficient to test FoCAL (see Supplementary File for more details). In this experimental setup for FoCAL on CORe-50, we combined all 8 training sessions in CORe-50 to generate 400 training object instances. In each increment, we randomly sampled $m = 5$ object instances (from 400 instances) and allowed the model to learn the label of $k = 1$ out of 5 object instances, making it a challenging problem to learn from a single object instance in each increment. Once an object was learned by the model, it was removed from the complete set of objects and, thus, was not available to the model in later increments. Hence, we allowed the model to learn all objects in 400 increments with one object learned in each increment. We used the test set accuracy at each increment as the evaluation metric. After training

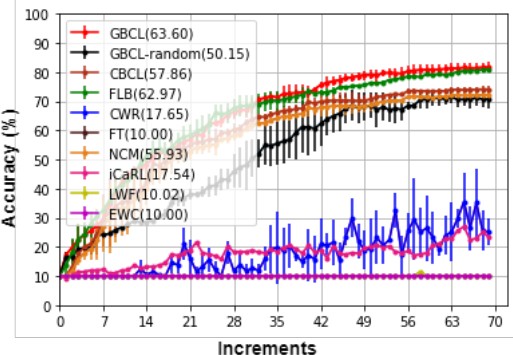

| Methods | Increments |
|---------|------------|
| FT | 23 |
| LWF | 23 |
| EWC | 24 |
| iCaRL | 18 |
| NCM | 17 |
| CWR | 21 |
| FLB | 19 |
| CBCL | 17 |
| GBCL-rand | 22 |
| **GBCL** | **15** |

Figure 2: Comparison of our method (red curve) to SOTA approaches in terms of classification accuracy on the Core-50 dataset. Average incremental accuracy is reported in parenthesis. The curves show average and standard deviation of 5 runs with random seeds.

Table 1: Total number of increments needed by each method to learn 10 classes in CORe-50.

in each increment, the model was tested on the complete test set of the CORe-50 dataset. We only report the accuracy for 70 increments for this experiment because all the approaches learn all 10 classes by 70 increments and the final accuracy starts to saturate.

For robustness, all the experiments were performed 5 times with random seeds. We report the average and standard deviation of the accuracies. Hyperparameters P and $\delta$ were chosen using cross-validation and were set to 0.2 and 0.7, respectively for all increments. For the shallow neural network used for classification in GBCL, we used a linear layer of the same size as the feature vectors generated by the ResNet ($512 \times 1$). The shallow network was trained for 25 epochs using the cross-entropy loss optimized with stochastic gradient descent (with 0.9 as momentum). A fixed learning rate of 0.01 and minibatches of size 64 were used for training.

## 3.2 Experiments on the CORe-50 Dataset

We compare our approach against 6 continual learning approaches (LWF [27], EWC [4], CWR [22], iCaRL [3], NCM [11], CBCL [10]), finetuning (FT) and a few-shot batch learning baseline (FLB) [28]. FLB uses a pre-trained CNN to extract feature vectors for images and trains a linear classifier using cross entropy loss. FLB is trained on the complete training data of the previous increments and the current increment. In other words, FLB does not learn continually and therefore should have an advantage over other continual learning approaches (including ours). FT uses the same architecture as FLB but FT is trained only on the data of the current increment. FT suffers from catastrophic forgetting and therefore should produce lower accuracy than other continual learning approaches. To the best of our knowledge, none of the eight approaches can be directly applied to FoCAL. Hence, for FLB, FT, LWF, EWC and CWR we used softmax-based uncertainty sampling to find most uncertain objects. For the softmax-based uncertainty scores, we found the softmax output of each image for an object instance. Then, we took the average of the maximum probability in the softmax output of each image of the object. The object with the highest average probability score was chosen to be labeled. Because iCaRL, NCM, and CBCL use centroids for classification, we used the average distance of the images of the new object from the centroids as the uncertainty score. A brief description of all six continual learning approaches is provided in Supplementary File.

### 3.2.1 Comparison with SOTA Approaches

Figure 2 compares our GBCL approach against random sampling and other SOTA approaches in terms of classification accuracy on the fixed test set for 10 classes. Our approach (GBCL) and FLB produce similar results for all 70 increments and GBCL produces slightly better accuracy than FLB in some increments. FLB, however, uses the data in all the previous increments at each new increment, while our approach only uses the data in the current increment and pseudo-feature vectors of the previous classes. We believe the reason for similar results for the two approaches is that GBCL generates and uses pseudo-feature vectors for previous classes to mitigate catastrophic forgetting,

| Increments | 40 | 80 | 120 | 160 | 200 | 240 |
|---|---|---|---|---|---|---|
| **Accuracy (%)** | 56.7 | 76.5 | 84.1 | 87.5 | 87.9 | 88.3 |
| **No. of Classes** | 18 | 20 | 20 | 20 | 20 | 20 |

Table 2: Test set accuracy and number of classes learned by Pepper over 240 increments.

and it also learns the most informative samples in each increment. As expected, FT suffers from catastrophic forgetting since it finetunes the network on new data in each increment. Regularization techniques (LWF and EWC) also suffer from catastrophic forgetting and produce the same accuracy as FT. CWR and iCaRL produce slightly better accuracy, however they also suffer from catastrophic forgetting. NCM produces much better accuracy than the other approaches with the help of centroids to remember past classes and prioritize objects to learn. However, NCM's accuracy is significantly inferior to FLB and GBCL. CBCL produces the best accuracy among the SOTA methods because it learns multiple centroids per class. However, CBCL's accuracy is still lower than GBCL.

Note that after 400 increments, FLB produces ∼89% final accuracy and GBCL produces ∼88% (1% lower) accuracy. The reason is that after 400 increments both models learned all the training samples in the dataset, and therefore the advantage of informative sampling in GBCL fades away. Further, unlike FLB, GBCL does not have access to all the data in the previous increments, and it suffers from slight forgetting compared to batch learning (FLB).

In terms of memory storage, GBCL stores 239 clusters/Gaussian distributions (centroids and covariance matrices) for all the classes. To avoid huge memory storage, we only use the diagonal entries of the covariance matrix for all the experiments. Therefore, GBCL stores 239×2=478 vectors of size 512 in memory after learning over 70 increments on the CORe-50 dataset, which requires only 0.97 MB. In contrast, FLB (batch learning approach) stores all 21000 features vectors for all the classes, which requires 43.08 MB (44 times more than GBCL). For other approaches, FT does not store any data, while LWF, EWC store minimal information, such as the predicted labels for the previous classes and fisher matrix to capture weight importance. CWR stores an extra weight matrix for the classification layer which requires 0.02 MB of storage space. iCaRL stores 2000 raw images for the previous classes, which requires 393 MB. NCM stores only 10 centroids for the previous classes, which requires 0.02 MB. Finally, CBCL stores 315 centroids which requires 0.64 MB. Note that all of the approaches also store a ResNet-18 model trained on the previous classes which requires 83 MB of space. This analysis shows that GBCL provides the best trade-off between memory storage and the overall performance in comparison with the other approaches.

### 3.2.2 Comparison with Random Sampling

To provide more insight into our approach, we tested our GBCL approach without active learning (GBCL-random). In this case, GBCL gets a random object to be labeled in each increment. GBCL-random produced significantly lower accuracy than FLB and GBCL for all the increments, especially in earlier increments. These results depict the contribution of our active learning approach to efficiently choose which objects to learn. In comparison with other SOTA approaches, GBCL-random beats all the approaches (except CBCL) in later increments but it is below NCM and CBCL in earlier increments. These results also demonstrate the effectiveness of GBCL (Section 2.1) of our approach to mitigate catastrophic forgetting.

### 3.2.3 Comparison of Learning Efficiency

To further evaluate the efficiency of of our approach for FoCAL, we report the total number of increments taken by all the approaches to learn all 10 classes (see Table 1). GBCL requires the least number of increments (15) to learn the 10 classes. FLB takes 19 increments to learn all the 10 classes because it uses the softmax based sampling. NCM and iCaRL also take similar number of increments because they use centroids to calculate the uncertainty scores but do not have access to all the data of the previous increments. CBCL is the closest to our approach and learns all the classes in 17 increments. FT, LWF and EWC all use softmax-based sampling without using any previous data, so they take a large number of increments to learn all 10 classes. Lastly, random sampling takes the most number of increments (except for FT, LWF, and EWC) to learn all the 10 classes. These results clearly show that our approach is the most efficient at finding the unknown objects and learning them in a small number of increments.

### 3.3 Experiment on the Pepper Robot

In this experiment we used the Pepper robot to learn 240 household objects belonging to 20 classes (12 objects per class). We performed the experiment in an indoor lab environment with objects placed at four different locations in the lab with different backgrounds. At each location we placed 5 different objects on a table (Figure 2 (a, b) in the supplementary file). For this setup, the robot first localized the 5 individual objects in the image. We use RetinaNet [29] pre-trained on MS COCO dataset [30] for object detection and localization. The robot then captures images of the individual objects from multiple viewpoints (see Figure 2 (c) in the supplementary file for an example). These images are used by the active learning module of GBCL to find the most informative object. The user (experimenter) then provides the class label of the chosen object to the robot using a keyboard. The newly labeled object is then used to update GBCL. Details about our complete system for the Pepper robot are in the Supplementary File.

This experiment tried to reproduce some of the challenges that might arise in realistic household environments. For example, a large number of objects were learned at different times, over multiple days (8 weeks), depicting a true lifelong learning robot in real-world environments. Different objects were used in the training and test sets. Moreover, the background of the objects was non-ideal since many parts of the background could be viewed as objects by the object detector. For example, consider Figure 2 in the supplementary file where the air-conditioner and the window could be viewed as objects by the object detector. Objects also varied in size and some were almost transparent. Further, the lighting conditions were not ideal. Some images were taken at night, while some during the day with different variations of sun light coming through the windows in the environment (see Figure 2 and Figure 6 for examples. More examples in the Supplementary File).

To evaluate this experiment, we created a test set of 60 objects (3 objects per class) that were different from objects in the training set. We tested Pepper's ability to classify the objects in the test set after 40, 80, 120, 160, 200, and 240 increments of learning new objects. We also report the number of object classes learned by Pepper after each of these increments (Table 2). After only 80 increments, Pepper learned all 20 object classes and achieved 76.5% accuracy on the fixed test set. For the rest of the increments the increase in accuracy was minimal, especially in later increments (120-240), similar to the results on the CORe-50 dataset. The reason is that in the earlier increments Pepper learned objects belonging to a large number of classes quickly which increased the overall accuracy. However, in the later increments Pepper only learned about more objects of the previously learned classes, thus the increase in overall accuracy was minimal. We should also note that even after 240 increments (same as the total number of training objects) Pepper only learned 197 objects. The reason is that the object localization module was not perfect, and it failed to detect many objects (43 objects).

## 4 Ablation Studies

We performed two ablation studies to examine the contribution of different components of our approach and the effect of hyperparameter values on GBCL's performance. This set of experiments were performed on the dataset collected by the Pepper robot (we call it Pepper dataset in this paper). As GBCL's accuracy saturates by 160 increments (Table 2), we performed all of these experiments for 160 increments, with a single object learned in each increment. We report accuracy for increments 0, 40, 60, 80, 120, 140, 160, and the average incremental accuracy. The rest of the experimental setup was the same as in the experiments for the CORe-50 dataset (Section 3.2).

For the first ablation study, we investigate the effect of hyperparameter $\delta$ on choosing the two acquisition functions (entropy and viewpoint consistency) for active learning. Figure 3 (a) depicts the impact of choosing different values ranging from 0 to 1, for the hyperparameter $\delta$. For the extreme values i.e. 0 and 1, our model chooses only one of the acquisition functions for active learning. Therefore, the model produces the lowest accuracy for $\delta$ values 0 and 1. However, there is no significant difference in accuracy when choosing either of the acquisition functions. For other $delta$ values close to 0 or 1, the accuracy increases slightly, but the best accuracy is achieved with $\delta = 0.7$. However, for other values of $\delta$ close to 0.7 (such as 0.6), there is not a significant difference in accuracy (only 0.07%), which shows that our approach is not highly sensitive to a large range of possible values of $\delta$. These results also confirm that choosing a combination of the two acquisition functions produces better performance than choosing either of them alone.

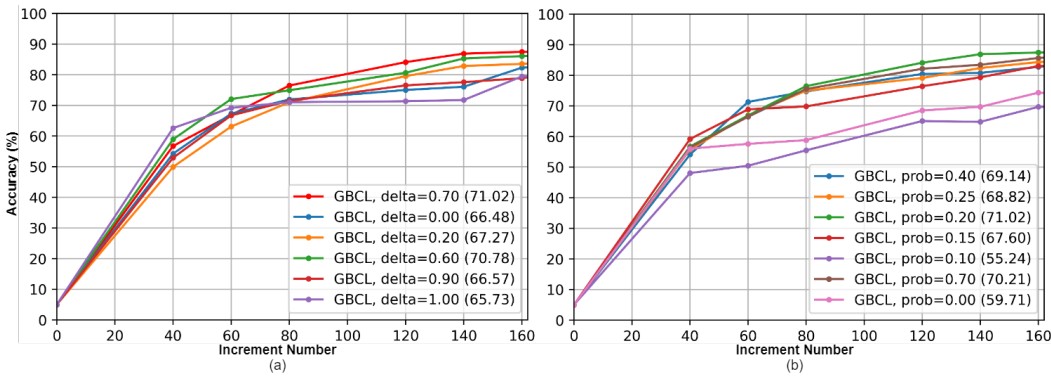

Figure 3: The effects of varying $\delta$, and the probability threshold $P$ on the classification accuracy on the test set of Pepper dataset. While changing one of the two parameters, the other parameter's values was set to be $\delta = 0.7$, $P = 0.2$ for the best results

For the second ablation study, we investigate the effect of hyperparameter $P$ that determines the number of clusters (or Gaussian distributions) generated for each object class. Figure 3 (b) shows the impact of $P$ on GBCL's performance. For $P = 0$, GBCL simply stores a single cluster for each class which might not be enough to capture the complex distributions of the object classes. Therefore, for $P$ values close to 0, there is a significant drop in GBCL's accuracy ($\sim$18%). As the $P$ value increases, the model starts to assign more clusters for each object class, and thus it is able to capture the complex distributions of the object data. The average incremental accuracy of the model increase significantly when $P = 0.15$ with 41 total clusters for all 20 object classes. The best accuracy is achieved for $P = 0.2$ (83 total clusters), however, for a large range of $P$ values the average incremental accuracy stays within a range of $\sim$3%. As the $P$ value continues to increase, the model starts to assign more clusters per class, therefore for higher values of $P$ the model uses more storage space. Further, for $P$ values close to 1, the model starts to recruit more clusters per class with only a few features (or a single feature) assigned to each cluster. Thus, the model starts to resemble more with the batch learning case when all the features of the previously learned classes can be stored. Because of this, we start to see an increase in accuracy again when $P = 0.7$ (379 clusters, 4.5 times more than the number of clusters stored for $P = 0.2$). Overall, these results show that our model is not highly sensitive to probability threshold values within a large range, and values close to $P = 0.2$ provide a nice trade-off between memory storage and model's performance.

Finally, note that we found both hyperparameters using cross-validation on the CORe-50 dataset and not on the Pepper dataset. However, even on the Pepper dataset the same hyperparameter values produce the best results. This shows that the chosen hyperparameter values are not dependent on a single dataset.

## 5 Related Work

Most continual learning approaches are designed for class-incremental setting in which a model (often a neural network) learns from training data of different classes in different increments and then evaluated on the test data of all the classes learned so far [3]. Most existing class-incremental learning (CIL) methods avoid catastrophic forgetting by storing a portion of the training samples from previous classes and retraining the model on a mixture of the stored data and new data [3, 31]. However, this approach does not scale as additional data exhausts memory capacity limiting performance in real-world applications. To avoid this problem, some CL approaches use regularization techniques [27, 4]. Although these approaches solve the memory storage issues, their performance is significantly inferior to approaches that store old class data. A few works have been proposed for FSCIL, to continually learn from a few-examples per class using limited memory [10, 9, 32, 33]. However, as mentioned in [17, 34], one significant limitation of all CL approaches is that they require the complete training data of each class to be labeled and available in a single increment, however in real-world robotic applications data is available in a streaming manner (online learning [19]) and it might be mostly unlabeled.

For object learning, many researchers have presented active learning techniques using uncertainty sampling [35, 18, 14, 15, 36, 16]. Most of these approaches train deep neural networks with special loss terms such that the networks can predict the most uncertain samples. All of these approaches, however, are trained for batch learning setting and will thus suffer from catastrophic forgetting when attempting to learn continually. Further, active learning approaches can predict unknown classes in the first increment after batch training. However, in a continual learning setting these approaches lose their ability to recognize unknown classes in subsequent increments because the model will assign the unknown classes to the newly learned classes (learned in the previous increments) that were unknown in the first increment.

Another related field to active learning is open-set recognition (or out of the distribution detection (OOD)) [37, 38, 38, 39], in which a model that is trained on a set of tasks/classes might face samples belonging previously unknown classes in the test set. Therefore, during the testing/deployment phase, the model must be able to detect which of the samples belong to unknown classes. Although methods for OOD [40, 41] have been developed to detect unknown classes, these models do not train on the newly encountered unknown samples. Mundt et al. [17] present a thorough analysis of previous continual and active learning, and OOD works, and suggest a common viewpoint with open-set recognition acting as a natural interface between continual and active learning. Towards this goal, we present an approach that allows an AI agent to actively select the most informative (unknown) samples from unlabeled data, and continually learn from the actively selected object samples.

## 6 Conclusion

This paper presents and evaluates a novel method for a challenging problem: Few-Shot Continual Active Learning (FoCAL) for object classification. Experimental results demonstrate that our approach for FoCAL is highly efficient and helps the model learn the most uncertain objects continually without forgetting earlier classes. Our approach not only outperforms the SOTA approaches on a benchmark dataset but also allows a robot to actively learn objects in a real environment over a long period of time. Finally, we have also released the dataset collected by the Pepper robot as a benchmark for future evaluations of FoCAL models.

The work presented here has a few limitations. 1) The experimental setup is still quite simple compared to a real household environment. 2) We assume that correct object labels are provided by the human assistant to the robot. 3) We did not recruit real participants to teach the robot, but the experimenter acted as a user for the robot. 4) We used a fixed feature extractor and object detector. In the future, we hope to work on these limitations and test our system with real human participants. Particularly, we hope to develop techniques that can allow feature representations to be learned from a few samples over a large number of increments.

There are a number of positive and negative long-term outcomes of this work. With respect to positive outcomes, the improved performance for continual learning by our approach may one day lead to robots that can tailor their behavior and learning to the needs of a person. We envision a home service robot [42] that uses continual learning to learn a person's food preferences before serving them breakfast, or a domestic service robot that learns how a person prefers their home to be cleaned and adjusts its behavior to their preferences. In terms of negative outcomes, one potential downside is that an active learning robot might learn unnecessary objects which are never going to be used by the robot. Learning such objects can potentially decrease the recognition performance on necessary objects that are required by the robot. Using FoCAL to learn unnecessary objects can also take valuable time that might be better used by the robot to perform necessary actions. For example, consider a dish washing robot which only needs to know kitchen utensils. The robot does not need to learn mechanical tools or children's toys but uncertainty-based sampling will cause the robot to learn these unnecessary objects. In such cases, some constraints might be added to a FoCAL system in order to avoid the robot from learning unnecessary objects.

## Acknowledgements

The authors acknowledge helpful comments of Alan R. Wagner for a preliminary version of this work.

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
