# OpenReview forum: "Few-Shot Continual Active Learning by a Robot"
_NeurIPS.cc/2022/Conference — NeurIPS 2022 Accept_

### Official Review · Reviewer_ACp7 · 2022-07-11

**Rating:** 6
**Confidence:** 3
**Soundness:** 3 good
**Presentation:** 3 good
**Contribution:** 3 good

**Summary:**

This paper proposed a new task combining few-shot learning, active learning, and continual learning, in the context of robots perceiving and interacting with unseen objects. In this task, new instances of different object categories emerge and the robot needs to identify the unseen ones and ask a human for labels. To tackle this problem, the paper proposed GMM Based Continual Learning (GBCL), where Gaussian mixture models are learned continually for each object, given a pretrained and fixed feature extractor. The confidence and consistency predicted by GMMs are used to select novel object instances for active learning. The experiments are performed on the CORe-50 dataset and on a real robot. The results demonstrate the effectiveness of the proposed method over baselines.


**Questions:**

* What is “GBCL-curiosity” in Line 240-241?
* How will a baseline that stores all feature vectors from all increments perform?
* Is the network doing classification on object-level or instance-level?


**Limitations:**

I appreciate the authors’ discussion on limitations in the conclusion section. In addition, I think the usage of a pretrained and fixed feature extractor and the detector is another major limitation of this paper.


**Strengths And Weaknesses:**

Strengths:
* The proposed task is very interesting and has a huge value for real-world applications of vision and robotic systems. Robots deployed in the real world need to interact with potentially unseen objects every day. This task is a combination of few-shot, active and continual learning. I think this paper formulates this task in a decent way. The experiments are performed both on a pure static dataset (CORe-50) and on real robots with human labeling in the loop.
* The application of GMM in continual and active learning in novel and the results prove to exceed the state-of-the-art.

Weaknesses:
* It’s unclear whether the comparison with other continual learning baselines is fair or not.
    * How is the memory usage of different continual learning? How many images or latent features are stored for each method? Which method stores a checkpoint of the network and which does not?
    * What’s the number of Gaussian mixtures during the learning of GBCL? How does this compare to the memory usage of other CL methods?
    * Are the networks for different CL methods all pretrained in the same way? During continual learning on CORE-50, do they all use the fixed feature extractor or the entire network is trained continually?
* The feature extractor for each object is pretrained and fixed. The object detector used in the robot experiment is also fixed. It is possible that the feature extractor is not capable enough and can achieve better performance if they are learning on new objects observed.
* It would be better if some baseline methods can be also tested on the robot experiment and compared to GBCL.

---

> ### Author Response · Authors · 2022-08-02
> **Comparison of memory usage added; new experiments on the Pepper dataset**
>
> We thank you for your insightful comments and have used these comments to improve the paper.
>
> Weaknesses:
>
> Memory Usage: We have added a discussion about the memory usage of all the approaches in the paper (L 258-271). In particular, GBCL requires only 0.97 MB of space to store GMMs of the previous classes. In contrast, FLB requires 43.08 MB (44 times more than GBCL). For other approaches, FT does not store any data, while LWF, EWC, and CWR store minimal information, and they all lead to catastrophic forgetting. iCaRL stores 2000 raw images for the previous classes, which requires 393 MB. Finally, NCM stores only 10 centroids for the previous classes, and CBCL stores 315 centroids which require 0.02 MB and 0.64 MB, respectively. All the approaches also store a ResNet-18 model trained on the previous classes which require an extra 83 MB of space. This analysis shows that GBCL provides the best trade-off between memory storage and overall performance in comparison with the other approaches.
>
> Network Pre-training: Yes, all approaches start with a ResNet-18 pre-trained on the ImageNet dataset. However, FT, FLB, CBCL, GBCL, CWR, and NCM use the pre-trained network as a frozen feature extractor, while iCaRL, LWF, and EWC finetune the pre-trained network on the new object data.
>
> Pre-trained Feature Extractor: Please see our answer to reviewer 6wg4 comments. Particularly, we have added a new experiment (Section 7 in the supplementary file) to compare our approach with FSCIL approaches that train the feature extractor for few-shot class incremental learning (FSCIL). The results show that GBCL is able to avoid drastic accuracy decreases over a large number of increments while using a pre-trained feature extractor. In contrast, FSCIL approaches that also update the feature extractor suffer from a steep decline in accuracy. These results show that it might be beneficial to keep the CNN feature extractor fixed when continually learning from smaller sample sizes over a large number of increments.
>
> Experiments on Pepper Dataset:
> Thank you for suggesting testing other methods on the Pepper dataset. We have tested some of the other approaches on our robot’s dataset and added the results in the supplementary file (Section 5). These results are similar to the results on the CORe-50 dataset, and all the approaches show the same trend in terms of overall accuracy. In particular, GBCL shows similar performance to the batch learning approach (FLB) that stores and retrains on all the feature vectors of the previous classes. We will release the dataset as a part of this paper so that future approaches can use it as a benchmark for FoCAL evaluations.
>
> Questions:
>
> • “GBCL curiosity” was a typo. We have fixed it in the paper.
>
> • The few-shot learning baseline (FLB) in our experiments stores all the feature vectors of the previous classes, and GBCL’s performance seems to be very close to this batch learning baseline (Figure 2 in the paper, Figure 4 in the supplementary file).
>
> • The network is doing classification on the object level.
>
> Limitations:
>
> Thank you for the helpful suggestion. We have added further discussion about the limitations of the pre-trained feature extractor and the detector to Section 5 of the paper.

---

### Official Review · Reviewer_6wg4 · 2022-07-11

**Rating:** 5
**Confidence:** 5
**Soundness:** 3 good
**Presentation:** 2 fair
**Contribution:** 2 fair

**Summary:**

This paper operates in the few-shot continual active learning (FoCAL) setting as exemplified by visual object category learning, but without concerning itself with continual representation learning. Instead, attention is mainly given to active, sample efficient learning of a classifier on top of pre-trained representations. A human-in-the-loop interaction model motivates the FoCAL setup, hence the paper does not only propose a solution for continual classifier learning, but it also motivates choices for the acquisition function used to select which samples to be labelled during the active continual learning process. Experimental validation is done on CORe-50 and a custom collected dataset using a robot.

**Questions:**

* Is there no need for further representation learning? Is such an assumption realistic?
* How are limitations of pre-trained representations addressed by the machinery on top?


**Limitations:**

Limitations should be discussed at length in the main text, not the appendix.

**Strengths And Weaknesses:**

Strengths:
* The paper identifies an important application of open-world learning in the particular case of visual concept learning with the constraints of a real-world robotic setup.
* The few-shot active classifier learning problem is interesting even outside of the continual learning paradigm.

Weaknesses:
* The paper does not focus on continual representation learning, since the feature extractor is pre-trained and fixed. This means reduced significance, since there exist several approaches for “shallow” continual learning, e.g. ELLA [Ruvolo et al. 2013].


FoCAL is not only about visual concept learning, but realistic robotic settings should also address issues of control, for which pre-training or hard-coding of behaviours are major limitations. The paper should limit claims to visual category classifier learning; this detracts from the clarity of the paper at the moment.

It would also help clarity if the work positioned itself better with respect to other continual learning works and made its assumptions explicit, see for context: [Hadsell et al. 2020, Mundt et al. 2020, 2022]

Several baselines [LWF, EWC, iCaRL] are designed for continual representation learning under different assumptions, and this should be discussed, also to improve clarity.

Few-shot meta-learning with pre-trained representations is well known to be effective [6], so it is difficult to argue for the novelty of the approach or the experimental setup.

It is difficult to argue for the high significance of the work due to not addressing open problems, such as continual representation learning. It is not surprising that pre-trained visual representations are useful in few-shot settings with novel visual categories, see the few-shot meta-learning works.

The paper should mention few-shot meta-learning works which have explored several of the concepts used, e.g. see the citations in [6]. There are too few references to works on open-world learning, a related field.



### References:

[Ruvolo et al. 2013] Paul Ruvolo, Eric Eaton. ELLA: An Efficient Lifelong Learning Algorithm. Proceedings of the 30th International Conference on Machine Learning, PMLR 28(1):507-515, 2013.

[Hadsell et al. 2020] Raia Hadsell, Dushyant Rao, Andrei A. Rusu, Razvan Pascanu. Embracing Change: Continual Learning in Deep Neural Networks. TiCS 2020.

[Mundt et al. 2020] Martin Mundt, Yong Won Hong, Iuliia Pliushch, and Visvanathan Ramesh. A Wholistic View of Continual Learning with Deep Neural Networks: Forgotten Lessons and the Bridge to Active and Open World Learning. CoRR 2020.

[Mundt et al. 2022] Martin Mundt, Steven Lang, Quentin Delfosse & Kristian Kersting. CLEVA-COMPASS: A CONTINUAL LEARNING EVALUATION ASSESSMENT COMPASS TO PROMOTE RESEARCH TRANSPARENCY AND COMPARABILITY. ICLR 2022.

---

> ### Author Response · Authors · 2022-08-02
> **Experiments added to test the effect of pre-trained feature extractor**
>
> We thank you for your insightful comments and have used these comments to improve the paper.
>
> Pre-trained Feature Extractor:
>
> One of the main limitations of the Few-Shot Learning (FSL) and Few-Shot Class Incremental Learning (FSCIL) approaches is that if a neural network is trained from scratch using only a few examples per class, it does not produce good accuracy. Therefore, for a few images/samples per class, FSL and FSCIL approaches first train the neural network model on a large number of base classes with a large number of images per class to learn a good feature representation. Some FSL approaches keep the feature extractor fixed [24], while others (such as few-shot meta-learning approaches) update the feature extractor as well when learning from a few classes. Most FSL and FSCIL approaches first train the CNN on a set of base classes from the same dataset as the one used in the few-shot phase. However, for the FoCAL setup, particularly for the robotics applications, it might not be possible to capture a large amount of data from the same environment where the system is deployed later on. In other words, we do not have the base classes for the same dataset in FoCAL. Therefore, in this work, we choose to use a generic feature extractor pre-trained on a large dataset, such as ImageNet. Further, it has been shown in [24] that when the base classes are coming from a different dataset (such as ImageNet) and the few-shot classes are from another dataset (such as CUBS dataset), the baseline approaches with the fixed feature extractor outperform the meta-learning approaches. Therefore, for FoCAL it might be best to use a fixed feature extractor.
>
> To further explore this point, we performed another experiment to compare GBCL against the FSCIL approach that also updates the feature representation when learning continually from a few classes (Section 7 in the supplementary file). As the FSCIL approach (termed TOPIC [6]) is not designed for FoCAL, we tested GBCL on the FSCIL setup described in [6] and removed the active learning component from GBCL (i.e. GBCL only used the GMM representation and pseudo-rehearsal and not the active learning techniques). For a fair comparison, we used the same settings as used by TOPIC [6] i.e. we trained a ResNet-18 on 60 classes in CIFAR-100 in the first increment and then learned the rest of the 40 classes over 8 increments with 5 classes per increment (5 images per class). However, unlike TOPIC, we do not continue to update the CNN in the 8 increments and use it as a fixed feature extractor after the first increment. As shown in Figure 6 in the supplementary file, GBCL produces a lower accuracy than TOPIC in the second increment only, because TOPIC adapts its feature representation to the new classes while GBCL uses the fixed feature representation. However, for the rest of the 7 increments, GBCL’s accuracy stays steady but TOPIC’s accuracy decreases drastically. The reason is that by adapting the feature representation to the new classes with only a few examples per class, the feature representation becomes too specific to the newly learned classes, which results in forgetting the previous classes. In contrast, GBCL continues to use the fixed feature representation learned from a large number of classes in the first increment and instead learns the complex distribution of new data in terms of GMMs. These results show that GBCL can address the limitations of the pre-trained feature representation by learning the complex distributions of the classes using GMMs and it can avoid forgetting using pseudo-rehearsal. Therefore, it can produce significantly higher accuracy (8\% higher) than FSCIL approaches that also learn the feature representation.
>
> Related Works:
>
> Thank you for suggesting related works [Hadsell et al. 2020, Mundt et al. 2020, 2022] that explored different directions for continual learning. We have added them to the paper and also positioned our paper with respect to these works (L 39-41, L356-360 in the paper). We have also added some works related to open-set recognition in the paper (L 351-355).
>
> Limitations/Discussion: As per your suggestion, we have moved the complete discussion from the supplementary to the main paper. However, we had to move Figures 2 and 4 (in the previous version of the paper) to the supplementary file (Figures 3 and 2 in the supplementary file) to create space for complete discussion in the paper. Finally, we have also added the pre-trained feature extractor as one of the limitations in the paper.

---

### Official Review · Reviewer_JcVg · 2022-07-11

**Rating:** 5
**Confidence:** 3
**Soundness:** 2 fair
**Presentation:** 3 good
**Contribution:** 2 fair

**Summary:**

This paper proposes a setup of few-shot continual active learning (foCAL), where the agent continually learns with an unlabeled data and limited amount of labeling budget. This problem setup is meaningful, because previous continual learning methods focused on the scenario in which the data of the current task are available and labeled. In contrast, the new setup assumes that the robot doesn't have access to the labeled data, and needs to ask human teacher for supervision with limited availability (hence, active learning). This paper proposes FoCAL for online learning for image classification task. The framework is well summarized in the Figure 1. The features extracted from unlabeled objects go through the acquisition function a(,) to get the most informative samples (k), which are labeled by oracle. These labeled samples from oracle are used to update GMM representations. Pseudorehearsal process is used to replay old data. In the final stage of classifier, the samples from old classes and labeled features are used. Framework design description is given, and the experiment validations are presented.

**Questions:**

Q1. I think the section 2.2. active learning using GMMs is the important part of the paper, and there are two essential techniques of this algorithm, namely (1) prediction entropy as acquisition function (2) viewpoint consistency. Is there any ablation studies for this? such as using just technique 1 in the first experiment, and just using technique 2 in the second experiment, comparing the performance with the algorithm.

Q2. Choice of hyperparameters. There are parameters P and \delta that affects the performance of the algorithm. Authors mentioned that P=0.2 and \delta=0.7 were chosen, and these values were chosen based on cross validation. Is there a graph or table that shows the performance of the GBCL algorithm based on different sets of those hyperparameters? How is the trend?


**Limitations:**

As authors mentioned, the main limitation of this work is that the experiments are quite limited and simple. It would've been a better paper if they had added more domain and more environments or different dataset. Some more experimental analysis on parameters and ablation studies would strengthen this paper.

**Strengths And Weaknesses:**

- Strength of this paper is the novelty of using GMM (Gaussian mixture model) based continual learning. The authors argue that the model system should recognize how different an incoming object (how new class it is) compared to other previously learned object classes with ease. To address this, they use a clustering-based approach; however, clustering-based approach normally uses mean feature vectors to represent object classes; instead, they use GMM to model each class data. Once the oracle gives label on the selected k feature vectors, the GBCL is applied to learn GMMs for the class. The updates of centroid is done by weighted mean of previous centroids and new inputs; the updates of covariance matrix is being done similarly.
- Another notable part of the paper is the section 2.2. active learning using GMMs. Especially, FoCAL employs two techniques for active learning. First technique is the use of prediction entropy as the acquisition function. Second technique is the use of viewpoint consistency; inconsistent predictions for the different view of the same objects generates a high reward for acquiring the label of the object, as opposed to the consistent predictions.
- Weak points: Is there an ablation studies for using these two techniques?
- Experiments and evaluations of FoCAL on Core-50 dataset, and evaluations on Pepper robot are presented.
- Comparison baselines are sufficient (FT, LWF, EWC, CWR, etc)

---

> ### Author Response · Authors · 2022-08-02
> **Ablation Studies Added**
>
> We thank you for your insightful comments and have used these comments to improve the paper.
>
> Ablation Studies:
>
> We have added the results of two different ablation studies to test the effects of hyperparameters $\delta$, and $P$ (Section 6 in the Supplementary File). We performed both ablation studies on the data collected by the Pepper robot, which contains 240 objects in the training set, and 60 objects (different from the training set) in the test set. (Note that the first suggestion to check the effect of using only one of the active learning techniques is covered in the ablation study for varying $\delta$)
>
> 1. For the first ablation study, we trained our GBCL model with different values of $\delta$ ranging from 0 to 1 (Figure 5 (a) in the supplementary file). $P$ was set to 0.2 in these experiments. The lowest accuracy is achieved when $\delta$ is 0 or 1 because the model uses only one of the active learning techniques. Although there does not seem to be any significant difference between the accuracy when using either of the two active learning techniques. For the other values of $\delta$, the model uses a combination of entropy and consistency scores. However, the best performance is achieved when $\delta$ is set to 0.7. Further, $\delta$ values close to 0.7 (i.e. 0.6) also achieve similar test accuracy. These results show that the best performance is achieved when using a combination of the two active learning techniques. Further, the accuracy of our approach is not highly sensitive to the choice of $\delta$ within a range of values close to 0.7.
> 2. For the second ablation study, we performed the same experiment on the Pepper dataset but changed the probability threshold for GBCL. $\delta$ was set to 0.7 in all of these experiments. Figure 5 (b) in the supplementary file shows the results of this ablation study. There is a significant drop in the model accuracy when using $P$ values close to 0. The reason is that for $P$=0 the model simply stores a single gaussian distribution to represent each class, which might not be sufficient to capture the complex distribution of the object classes. As the $P$ value increases, the model starts to assign more mixture components (gaussian distributions) to each class to capture the complex relationship between the classes. For $P$=0.2, the model achieves the highest accuracy and uses 83 total gaussian distributions to represent the 20 object classes. Variations of the $P$ value by a small range do not impact the accuracy significantly (less than a 3% decrease). However, for values higher than 0.2, the model starts to generate more distributions for the classes thus requiring more memory. And as the threshold increases to 0.7, the accuracy starts to increase again. The reason is that with the increase in the $P$ value, GBCL starts to recruit a large number of clusters for each class, with a small number of images per cluster, and thus it gets closer and closer to the batch learning case when the model can store all the feature vectors separately for each class (when $P$=1). Therefore, with $P$=0.2 our model gets a good tradeoff between memory storage and accuracy.
>
> Both ablation studies showed the contributions of different components of our model and also confirmed that GBCL is relatively insensitive to the two hyperparameters over a large range of possible values. Finally, note that we found both hyperparameters using cross-validation on the CORe-50 dataset and not on our dataset. However, even on our dataset the same hyperparameter values produce the best results. This shows that the chosen hyperparameter values are not dependent on a single dataset.

---

### Meta-Review · Area_Chair_uLx6 · 2022-09-10

**Recommendation:** Accept
**Confidence:** Certain

**Metareview:**

All reviewers appreciated the importance of the problem being tackled, and the effectiveness of the proposed method. There were a number of concerns about ablations and use of pre-trained feature extractors, but these have been sufficiently addressed in the authors' rebuttal. I agree with the reviewers in recommending acceptance.

**Award:**

No

---

### Decision · Program_Chairs · 2022-09-14

Accept